# Radar-Camera Fusion Network for Depth Estimation in Structured Driving Scenes

**DOI:** 10.3390/s23177560

**Published:** 2023-08-31

**Authors:** Shuguang Li, Jiafu Yan, Haoran Chen, Ke Zheng

**Affiliations:** 1School of Automation Engineering, University of Electronic Science and Technology of China, Chengdu 611731, China; 2School of Mechanical and Electrical Engineering, University of Electronic Science and Technology of China, Chengdu 611731, China

**Keywords:** depth estimation, radar, camera, dual-branch network

## Abstract

Depth estimation is an important part of the perception system in autonomous driving. Current studies often reconstruct dense depth maps from RGB images and sparse depth maps obtained from other sensors. However, existing methods often pay insufficient attention to latent semantic information. Considering the highly structured characteristics of driving scenes, we propose a dual-branch network to predict dense depth maps by fusing radar and RGB images. The driving scene is divided into three parts in the proposed architecture, each predicting a depth map, which is finally merged into one by implementing the fusion strategy in order to make full use of the potential semantic information in the driving scene. In addition, a variant L1 loss function is applied in the training phase, directing the network to focus more on those areas of interest when driving. Our proposed method is evaluated on the nuScenes dataset. Experiments demonstrate its effectiveness in comparison with previous state of the art methods.

## 1. Introduction

In recent years, the automotive industry is in an era of change, the rapid development of automated driving-related technologies, Google, Huawei, Baidu, Tesla and other companies are competing to participate in the research, and constantly promote the development of related technologies. Automatic driving technology is a comprehensive subject that crosses many disciplines, and its research content can be roughly divided into three sub-modules: perception, decision-making and control [1]. Among them, the perception system, as the “eyes” of the automatic driving system, is responsible for collecting road information and processing it to provide the necessary information for subsequent decision-making, and its importance is self-evident. Depth estimation is aimed at estimating the depth information in the scene through image information, which can effectively facilitate the realization of 3D target detection [2,3,4], scene reconstruction [5,6,7] and other tasks [8], and has extremely important application value for automatic driving.

Accurate depth estimation, which aims to predict the depth value of each pixel from the RGB image, can effectively guarantee the safety of autonomous vehicles. At present, as convolutional neural networks (CNN) have achieved great success in many other fields, it is also introduced into depth estimation tasks extensively, to predict the dense depth map corresponding to the input RGB image in an end-to-end manner, and it indeed shows its strong capability [9,10,11]. Owing to the use of CNN, the existing algorithms have greatly improved the accuracy of depth estimation compared to conventional methods. It is still a common paradigm for addressing this problem today. Many different types of networks and loss functions are proposed successively in order to pursue better performance of depth estimation.

However, monocular depth estimation has been shown to be an ill-posed problem. By introducing LiDAR to obtain additional depth information of individual pixels in the image, the depth estimation task is converted into the depth completion task, which can further improve accuracy [12,13,14]. The recent mainstream scheme is to leverage CNN to fuse multi-sensors data, showing remarkable progress in depth estimation. The surrounding scene can be perceived by different sensors that have their own characteristics. Cameras are low cost and provide high-resolution images with rich texture and color information. LiDAR can provide accurate depth information but is limited by cost and computing power, making it difficult to deploy under constrained conditions. Both sensors are difficult to operate in bad weather. In contrast, radar can capture the distance and speed of a target as a sensor that is not susceptible to bad weather but at a relatively low resolution. In this paper, we use radar and RGB images as inputs to our proposed architecture.

Unlike indoor depth estimation, the outdoor scene of autonomous driving is relatively structured, which has some inherent features. Despite the impressive success of existing methods, many of them almost ignore the above point. Intuitively, it is natural to seek to exploit these attractive characteristics and try to mine the latent semantic information of the image, which is considered prior knowledge to act on the network so that it can facilitate a depth estimation task. Specifically, design a novel dual-branch radar-camera network based on CNN, making the different parts in an image can be predicted respectively with the following fusion strategy, as shown in Figure 1. At the same time, we want to force the network to pay more attention to regions that are of interest to human drivers, such as roads, pedestrians, vehicles, etc., to obtain more accurate results for these regions while maintaining global accuracy. The contributions of this paper can be summarized as follows:A triple decoder architecture based on CNN is introduced into the network to address different areas of an image, which makes better use of latent semantic information in autonomous driving scenes;We apply a variant of the L1 loss function during the training phase to make the network more focused on the main vision objectives, which is more in line with human driving habits;We evaluate our proposed depth estimation network on the nuScenes dataset, showing that our approach can significantly reduce estimation error, especially in areas of greater interest to drivers.

## 2. Related Works

### 2.1. Fusion of Radar and Camera Applications

Radar technology is known for its high-precision obstacle detection capability, which can detect a wide range of targets such as vehicles, pedestrians, and buildings. Unlike cameras, radar operates using electromagnetic waves and is therefore unaffected by low-light conditions (e.g., sunset, nighttime, or inclement weather), enabling it to operate reliably in a wide range of environments, significantly improving the reliability of autonomous driving systems.

Fusing radar with sensors, such as cameras, not only provides richer information, but also enables the system to understand the surrounding environment more accurately. Radar–camera fusion has been widely used in many fields, such as security, earthquake relief and autonomous driving. In literature [15,16], the fusion of target velocity and azimuth information obtained by radar with image information not only ensures consistency of target tracking but also improves tracking accuracy. It is shown that the fusion of camera and radar is not only effective for target detection, but also plays a role in the fields of gesture recognition [17], vital signs monitoring [18], human bone detection [19] and earthquake rescue [20], among others.

Especially in the field of autonomous driving, the fusion of radar and camera is of great importance. For example, Hussain et al. [21] designed a low-cost method for detecting drivable areas in long-distance areas of self-driving cars by fusing radar and camera. Similarly, Wu et al. [22] solved the challenge of missing parking boundaries on maps or difficult parking spot detection by jointly using radar and cameras. In addition to sensing the external environment, Kubo et al. [23] proposed a non-contact driver drowsiness detection method that estimates driver drowsiness with high accuracy and detail. De et al. [24] estimated the vehicle’s position based on the fusion of radar and camera sensors, speed and direction information.

In summary, the fusion of radar and camera plays a key role in multidisciplinary applications, providing more accurate and comprehensive information by integrating the advantages of different sensors and promoting the development of automation technology in various fields.

### 2.2. Monocular Depth Estimation

Monocular depth estimation is quite a challenging subject, as 3D information is lost when images are fetched by a monocular camera. Traditional algorithms rely heavily on hand-crafted features, such as texture and geometry, combined with a probabilistic model [25,26]. Over the past few years, CNN has achieved a convincing effect on image processing, so it is introduced in depth estimation and gradually becomes the most popular method for this task.

In general, depth estimation is treated as a regression problem. Eigen et al. [27] construct a multi-scale deep CNN to generate a dense depth map for the first time. Some methods attempt to combine CNN with conditional random field (CRF) to improve network performance [28], but they also increase the complexity of the system. New architectures are proposed to better extract features. Laina et al. [29] design a fully convolutional residual network (FCRN) for depth estimation, which is an encoder–decoder structure. In this structure, the FC layer and the last pooling layer are no longer used, instead of an efficient decoder structure consisting of a series of upper convolutional modules, which significantly reduces the number of parameters. Some inspired improvements like [30] have been proposed since then. Some methods improve the model by introducing attention mechanisms [31,32]. Chen et al. [33] used a novel spatial attention block to guide different feature layers to focus on different structural information, i.e., local details or global structures. On the other hand, many researchers treat depth estimation as a classification problem, which divides the depth value into discrete bins where each pixel falls into one of them, followed by some post-processing means to map to the continuous space and obtain the final continuous depth map. Typical methods include Cao et al. [34], where CNN was used to determine the appropriate bin that pixels should fall into, and then a fully connected CRF was employed to generate continuous prediction maps from the derived classification result. Fu et al. [35] discretized the depth values in log space and treated it as an ordinal regression problem.

### 2.3. Depth Completion

Depth completion uses RGB images to give sparse depth maps for densification. Two typical challenges are how to better handle sparse data and how to integrate two modalities of data in an efficient manner.

Many methods based on CNN architecture have been proposed so far. As in Figure 2, early fusion models concatenate a sparse depth map directly with an RGB image and then send it to a network as a whole. Because the initial depth map is highly sparse, conventional convolutional operations have a poor effect on processing. Thus, Uhrig et al. [36] proposed a sparse invariant convolution that uses a binary validity mask to adapt sparse input. For the late fusion strategy, a dual-branch encoder is universal. The features of RGB image and sparse depth map are usually extracted, fused at intermediate layers and transmitted to the decoder in the following steps [37]; otherwise, depth maps are inferred by the respective features and the final map is obtained by merging two outputs [38]. Some work devises a two-stage network, simply predicting a coarse depth map in the first stage before passing it to the fine-tuning stage. Cheng et al. [39] proposed the convolutional spatial propagation network (CSPN), which refines the results by learning the affinity matrix and propagating sparse depth samples in local neighbors.

### 2.4. Depth Estimation with Semantic Information

The depth information in an image describes the spatial relationship of the scene, while the semantic information represents the physical nature of the scene. The two share similar contextual information. Therefore, it is natural for researchers to consider introducing semantic information into the task of depth estimation as an assistant. Typically, depth estimation and semantic segmentation are combined for training, and the parameters of two tasks are shared by a unified structure to promote each other. Zhang et al. [40] proposed a joint learning framework to recursively refine the results of two tasks. Zhu et al. [41] used semantic segmentation to smooth the depth quality of object edge regions.

The above studies show that the interaction between semantics and depth actually promotes the accuracy of depth estimation. Inspired by this, we consider the full use of semantic information in the structured scene of autonomous driving. The focus of our method is not to make the network achieve good results for both tasks, but to mine the semantic information of the scene and leverage it in an explicit way to facilitate our focused depth estimation task, making it more accuracy.

## 3. Methodology

### 3.1. Overview Architecture

As shown in Figure 3, we propose an end-to-end depth estimation network using an encoder–decoder architecture. The dual-branch encoder includes a branch with RGB images as input and a branch with sparse depth maps generated by radar as input, aiming to make full use of two types of data to achieve effective feature extraction. The two branches work independently without interference from each other. Specifically, the RGB-branch is constructed by ResNet-34 [42], which is pre-trained on ImageNet [43], containing four convolutional modules to gradually extract image features and obtain a feature map of 1/32 size of the original image. The situation is somewhat different for the radar branch, as the data are sparser compared to RGB images. This branch is built in two stages. The sparse depth map is first preprocessed by a series of sparse invariant convolutions before the derived features are transmitted to subsequent residual blocks for further feature extraction. Residual blocks are constructed symmetrically to the RGB branch, except that they contain fewer channels. The fusion of features takes place in the intermediate part of the network, where feature maps generated by two branches are concatenated and fed into the decoder.

### 3.2. Feature Extraction

In the image branch, we made the necessary adjustments to the ResNet-34 network to meet the requirements of feature extraction in the depth estimation task. The fully connected layer was deliberately removed. This structural adjustment can be clearly seen in Figure 4. After architectural modifications, the network takes the form of four key convolutional modules, each of which plays an important role in the feature extraction process. We perform a series of layer-by-layer convolutional operations to enable the image to progressively capture feature information at different levels of abstraction. This design enables the network to provide richer semantic information for the input image data, laying a solid foundation for subsequent task processing.

For the radar branch, the encoder has been constructed with special design considerations. This is due to the higher sparsity of radar data compared with LiDAR. As shown in Figure 5, the ratio of LiDAR to radar data projection points on the image plane is approximately 3000 to 100. However, conventional lidar input modules are not suitable for processing radar data.

Therefore, in this paper, we adopt a two-stage processing strategy for radar branch data. First, we pass the sparse depth image through a sparse invariant convolution block for initial feature extraction. Subsequently, we use the same residual blocks as in the image branch, except that the number of convolution channels is different, to further extract feature information about the radar data. This strategy allows us to obtain valuable information from sparse radar data more efficiently and lays a solid foundation for feature extraction for subsequent tasks.

As mentioned above, sparse invariant convolutional is used in the initial radar-branch processing. Firstly, we briefly review this approach, as shown in Figure 6, this convolutional operation introduces the validity mask corresponding to the sparse depth map as an additional input and passes the poison layer by layer, which is computed as
(1)$$f_{u,v}\left(x,o\right)=\frac{\sum_{i,j=-k}^{k}o_{u+i,v+j}x_{u+i,v+j}w_{i,j}}{\sum_{i,j=-k}^{k}o_{u+i,v+j}+\varepsilon }+b$$
where *x* is the sparse input, *o* represents the binary validity mask, which values 1 (observed value) or 0 (no observed value corresponding to input *x*), and ε > 0 is a small number to avoid division by 0 caused by no observed value in the convolution region. The subsequent mask is determined by the following formula:(2)$$f_{u,v}^{o}\left(x\right)=\max_{i,j=-k,\ldots ,k}o_{u+i,v+j}$$

As shown in Figure 7, we construct a structure to stack five sparse invariant convolutional layers, in which the output of the fourth layer is passed into subsequent residual blocks, and the output of the last layer also generates a feature map after up-sampling. Supervision is applied for better optimization.

### 3.3. Depth Decoder

For the architecture of the decoder, it can be clearly seen that the networks use a triple encoder structure. The importance and inherent characteristics of different regions vary greatly, so pixels in a highly structured driving scene are divided into three categories, namely sky; trees and buildings; road regions, pedestrians and vehicles. For simplicity, the three decoder branches are called sky-branch, tree-branch and road-branch. They are designed to predict depth values for each of these categories. We hope that the extracted features can be carried out to restore a depth map of the corresponding regions in each branch, derive more accurate results for three categories and thus improve the overall depth maps. As for the details of the decoders, all of them share the same structure, which is composed of four up-projection blocks [44], followed by a 3 ∗ 3 convolution and bilinear up-sampling operation; thus, the resolution is restored to the same as that of the original RGB image. The three depth maps will be merged in the following steps.

As shown in Figure 8, the up-sampling module inverts the pooling of the input image by 2 × 2 size, i.e., each pixel of the input is expanded into a matrix of 2 × 2 size, and the value of the pixel is placed in the upper-left corner of the matrix, and the rest of the pixel is filled with 0. After that, the inverted pooled feature map matrix is inputted into a module with a residual connection, which is divided into two parts, the upper branch and the lower branch, and the upper branch goes through a 5 × 5 size convolutional layer with step size 1 and filling with 2. The lower branch passes through a 5 × 5 convolutional layer with step size 1 and padding of 2. The upper branch first passes through a 5 × 5 convolutional layer with step size 1 and padding of 2, and then passes through a 3 × 3 convolutional layer with step size 1 and padding of 1 after ReLU activation.

### 3.4. Fusion of Depth Maps

We obtained three depth maps as preliminary results with the decoder. It is then necessary to fuse them together to obtain the final depth map. Three methods are applied in this paper. The easiest way is just to add them. In addition to this, we propose using confidence maps to fuse the results of triple-branch predictions. When each decoder branch predicts a depth map, the corresponding confidence map is also generated to perform the weighted sum operation. In order to provide semantic information to individual branches, as shown in the dotted line box in Figure 3, an additional decoder for semantic segmentation is introduced, which is similar to other depth decoders, except that the final output generates a segmentation map with 19 categories.

We further classify them into three categories as described above. Based on this, areas corresponding to each category in the three maps are selected and integrated into the final depth map, as shown in Figure 9c. Our fusion method can be formulated as:(3)$$Map_{add}=Map_{1}+Map_{2}+Map_{3}$$
(4)$$Map_{conf}=Map_{1}*Conf_{1}+Map_{2}*Conf_{2}+Map_{3}*Conf_{3}$$
(5)$$Map_{seg}=Map_{1}*Mask_{1}+Map_{2}*Mask_{2}+Map_{3}*Mask_{3}$$

### 3.5. Loss Function

The driver’s focus while driving is usually on the road, pedestrians or vehicles. In view of this, a more appropriate loss function is needed to optimize the model, so that network attention is more focused on these regions, which may be more closely similar to human driving habits. Thus, we use a loss function with weights as follows:(6)$$L_{depth}=\frac{1}{m}\left(\omega \cdot \sum_{\left(p,q\right)\in S_{1}}\left|d\left(p,q\right)-\hat{d}\left(p,q\right)\right|+\left(2-\omega \right)\cdot \sum_{\left(p,q\right)\in S_{2}}\left|d\left(p,q\right)-\hat{d}\left(p,q\right)\right|\right)$$
where *d* and $\hat{d}$ denote the ground truth depth map and the predicted depth map, respectively. $S_{1}$ denotes the regions of interest, i.e., the same class as the road-branch, $S_{2}$ denotes the other regions, i.e., the same class as the tree-branch and sky-branch, *m* is the number of valid pixels (not every pixel exist a ground truth value), ω is a parameter to be adjusted. We want to find a parameter that will balance the two.

In addition to calculating the above depth loss, we also add the smoothing term. Since depth discontinuities usually occur at the borders, the image gradient is used for weighting, defined as:(7)$$L_{smooth}=e^{-\left|\partial_{x}\left(I\right)\right|}\left|\partial_{x}\left(\hat{d}\right)\right|+e^{-\left|\partial_{y}\left(I\right)\right|}\left|\partial_{y}\left(\hat{d}\right)\right|$$
where $\partial_{x}$ and $\partial_{y}$ denote the gradient along the *x* and *y* directions.

In the training phase, the feature map generated by sparse invariant convolution blocks is also supervised as $L_{sparse}$:(8)$$L_{sparse}=\frac{1}{m}\sum_{i=1}^{m}\left|d_{i}-{\hat{d}}_{si}\right|$$
where *d* denotes the ground depth map. $\hat{d}s$ denotes the predicted sparse depth value in Figure 7.

Finally, when the depth maps are fused as shown in Figure 9c; we also need to monitor segmentation results. The segmentation loss function:(9)$$L_{seg}=-\frac{1}{N}\sum_{i=1}^{N}\sum_{j=1}^{C}Y_{i,j}log\left(P_{ij}\right)$$

Thus, the overall loss function is:(10)$$L_{total}=\lambda_{1}\left(L_{depth}+\lambda_{2}L_{smooth}+\lambda_{3}L_{sparse}\right)+L_{seg}$$
where $\lambda_{1},\lambda_{2},\lambda_{3}$ are hyperparameters.In this paper, we set $\lambda_{1},\lambda_{2},\lambda_{3}$ to 0.5, 0.001, and 0.3, respectively.

## 4. Experiments

### 4.1. Experimental Setup

Dataset: The nuScenes dataset [45] is a large open autonomous driving dataset that contains radar data. Each scene lasts 20 s with 40 key frames, and the resolution of each frame is 1600 ∗ 900. Our experiment used 850 of 1000 scenarios, 810 scenarios for training and 40 scenarios for evaluation (i.e., 32,564 images for training and 1585 images for evaluation). Interpolation and filtered LiDAR projections were used as supervisory data. All the inputs and labels used for the experiments in this paper are shown in Figure 5.

Training details: Our proposed method is implemented in PyTorch and uses the ADAM optimizer. Our training process lasts 30 epochs with an initial learning rate of 0.0005, which decays by 0.1 every 5 epochs, and the batch size is set to 4. In addition, color jitter and horizontal flip transformation are adopted as data augmentation. Hyperparameters in loss functions are set to $\lambda_{1}=0.5,\lambda_{2}=0.001,\lambda_{3}=0.3$. The network is trained on NVIDIA GeForce GTX TITAN X.

Evaluation Metrics: We apply the following standard metrics for evaluation:

Mean Absolute Error (MAE): $\frac{1}{n}\sum_{i=1}^{n}\left|d_{i}-{\hat{d}}_{i}\right|$

Root Mean Square Error (RMSE): $\sqrt{\frac{1}{n}\sum_{i=1}^{n}{\left(d_{i}-{\hat{d}}_{i}\right)}^{2}}$

Mean Absolute Relative Error (REL): 1n∑i=1n|di−d^i|d^i

Mean Absolute Relative Error (REL): 1n∑i=1n|di−d^i|d^i

Threshold Accuracy ($\delta_{i}$): % of $d_{i}$ s.t.$\max\left(\frac{d_{i}}{{\hat{d}}_{i}},\frac{{\hat{d}}_{i}}{d_{i}}\right)<thr,$ where $thr\in 1.25,1.25^{2},1.25^{3}$.

Where the depth value is calculated in meters, in addition to the above metrics, there are also metrics for evaluating the number of parameters and the inference speed of the model; the number of parameters is the total number of parameters that need to be trained in the training of the model. Inference time is the number of images and radar data the model can process per second.

### 4.2. Comparing Results

Our proposed method has been compared with previous methods on the nuScenes dataset, showing its competitive performance. The results can be seen in Table 1. The methods of Ma et al. [46] and Hu et al. [38] are designed for Lidar, encountering some incompatibilities in processing radar data. The radar-based method [47] achieves slightly worse results than our add and conf fusion methods, but far less than our seg fusion method, which improves RMSE by 0.373 and MAE by 0.216, demonstrating the effectiveness of our method.

### 4.3. Optimal Parameter ω of Loss Function

In order to determine the optimal parameters for the depth loss function, we used a series of experiments. We systematically tuned ω to a starting value of 0 and gradually increased it in steps of 0.2 until it reached 2. This way of varying the parameters allowed us to gradually expand the coverage of the region of interest, thus providing a comprehensive view of the performance of the loss function under different parameters.

For each value of the parameter ω, we conducted a series of experiments and recorded the corresponding results. These results include the error profile under different categories. These error data are organized and listed in Table 2, through which we can clearly observe the trend and distribution of errors in each category under different parameter settings. These results provide strong support for us to deeply explore the relationship between the performance of the depth loss function and parameter settings, and also lay a solid foundation for further analysis and discussion.

From the experimental results, it is clear that when the parameter ω is taken to a value of 1.4, we can obtain a scenario with the highest performance in this scenario. The lowest level of error was achieved in the study area, i.e., the road area. It is worth emphasizing that with this parameter configuration, the prediction accuracy for the other two regions was also maintained within acceptable limits. It is encouraging to note that even for the high contrast tree category, the accuracy has reached its peak.

However, with the gradual increase in the parameter ω, i.e., the gradual increase in the weights of the road category, we cannot ignore the sharp increase in the error rate of the road classification. Perhaps this can be attributed to over-attention to road categories, resulting in the loss of some key global information in the model. This bias may have disrupted the balance between different categories, which in turn affected the overall classification performance. This point deserves to be further explored and addressed in future studies.

### 4.4. Fusion of Depth Maps

To evaluate whether our method works, we set up a double encoder–single decoder network for comparison, which uses the exact same depth decoder as in our proposed method. At the same time, we train on three different fusion approaches to explore which would work best.

The experimental results are shown in Table 3. Compared to the single decoder network, the three different fusion modes all improve accuracy, which clearly shows that our model plays a role in depth estimation. In addition, the method with a seg decoder achieves the highest accuracy. It is possible to use semantic information directly, allowing the corresponding categories to be more specific.

To ensure the comprehensiveness of the study and a deeper understanding of the performance of the proposed method, a series of comparative experiments were conducted focusing on errors in the depth estimation at different distance ranges. This fine-grained comparative study aims to reveal the applicability and limitations of the proposed method from multiple dimensions. By comparing the methods in different distance contexts, we can gain a more accurate insight into the behavioral performance of the methods in depth estimation.

In the experiments, we focus our study on the error of depth estimation over different distance ranges to comprehensively capture performance variations at different depth intervals. This nuanced exploration helps us gain insight into whether the proposed method exhibits different performance trends under different distance scenarios. The experimental results are presented exhaustively in Table 4, which clearly presents the error data for depth estimation at each distance interval. This provides us with comprehensive information that allows us to gain a deeper understanding of the performance characteristics of the proposed method at different depth levels. Through this rigorous comparative study, we can more accurately assess the strengths and weaknesses of the proposed method and provide valuable insights for future research and improvement.

By analyzing the data in Table 4, we can clearly observe that the error in the depth estimation was greater in the far range. This phenomenon can be attributed to the fact that the depth distribution in the labeled data was mainly concentrated in the near and middle ranges, resulting in an imbalance in the distribution between categories. This imbalance makes it difficult for the neural network to obtain enough information to support long-range feature learning. To solve this problem, we adopt an optimal segmentation fusion strategy. Compared to the benchmark single-decoder model, we achieved an improvement of 3.95%, 5.98%, 6.69% and 11.59% in each distance interval from near to far, respectively. This improvement yielded significant results in all distance ranges, particularly in the accuracy of predictions in mid-range and long-range ranges. This also provides solid evidence for the effectiveness of our designed network on depth estimation tasks.

In addition, it is worth noting that this branch not only performs well in depth estimation tasks, but also has significant potential in mining semantically relevant features, thus providing further opportunities to optimize the performance of other branches. We clearly demonstrate the significant effectiveness of this branch at the level of semantic features through detailed experimental analysis, as well as case-specific visualization of Figure 10. We can intuitively perceive the branch’s excellent performance in capturing semantic information that is closely related to the image content, which further enhances the network’s sensitivity to individual features of the image. At the same time, this effective mining of semantic information plays a positive role in improving the performance of other branches, further highlighting the important value and multifaceted impact of this branch. This helps to further confirm the overall superiority of our proposed network architecture and its effectiveness at different levels.

After an exhaustive visual presentation, we further validate the significant improvement of the method proposed in this paper in depth estimation tasks with the example images shown in Figure 10. In the first row of images, we can observe that in the single decoder model, the car on the left side of the image is mixed with the background, losing the clarity of the car outline. However, our proposed model can recover the contour information of the car in this scenario. This side-by-side observation confirms the excellent performance of our model in preserving object details.

In addition, it is worth emphasizing that the network designed in this paper shows superior performance in the depth estimation of small-scale targets, such as trees, utility poles, and other objects. At the same time, our method also shows excellent ability to recover the local detail structure of the image. It is able to predict depth values more smoothly, while at certain depth discontinuities and object edges, the network is able to clearly render depth changes, compared to the blurrier performance of a single decoder model at these locations. This series of observations further confirms the effectiveness of our proposed method, especially its superiority in handling image details, edges and small objects, providing solid support for the application of our method in real-world scenarios.

## 5. Conclusions

In this paper, we present a convolutional neural network architecture specifically designed to perform depth estimation, especially for structured driving scenarios. Our network model processes image and radar data in a more systematic way, and then feeds the extracted features into three independent decoders for predicting different classes of depth information separately. With this approach, we use multiple strategies to fuse depth maps. Also, in order to focus the network on regions that are more interesting to human drivers, we employ two classes of weighted depth loss functions. The effectiveness of our approach is confirmed by several experimental proofs.

In recent years, depth estimation methods based on deep learning have attracted a lot of attention. Although our proposed method has achieved some improvement in prediction accuracy, it still has some limitations. For example, the network does not yet perform well in predicting depth values for long-range targets and edge regions. In addition, the network model has more parameters, which limits its real-time performance. To address these issues, we encourage future research in the following areas: further improvement of the network to enhance depth estimation of distant targets and edge regions, and optimization of the network structure to reduce model parameters to enhance real-time performance. These efforts will help advance the development of deep learning in the field of depth estimation.

## Figures and Tables

**Figure 1 sensors-23-07560-f001:**
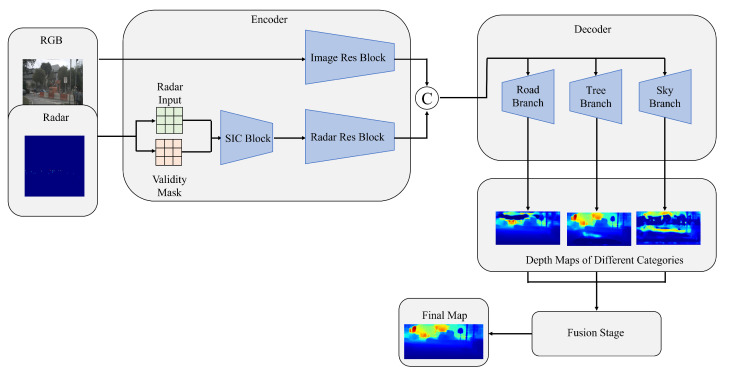
We try to predict different parts of an image using different decoders, expecting it to extract potential semantic information. SIC Block represents the sparse invariant convolution.

**Figure 2 sensors-23-07560-f002:**
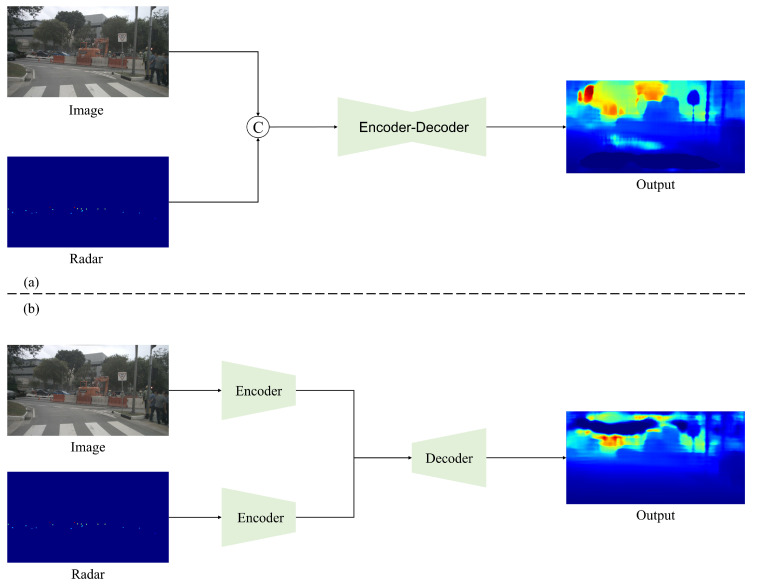
Different stages of fusion. (**a**) Early fusion; (**b**) late fusion.

**Figure 3 sensors-23-07560-f003:**
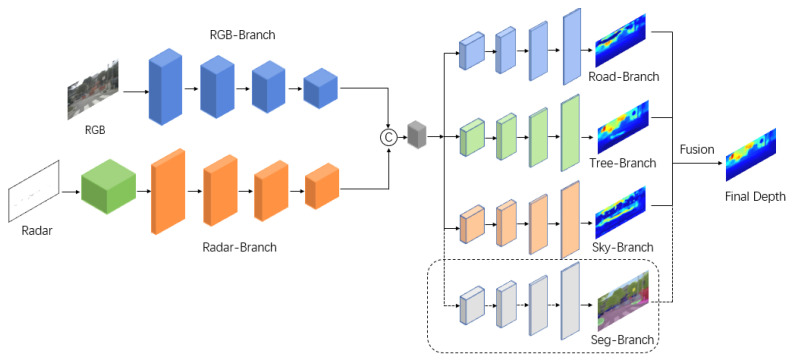
The architecture of the proposed method. We use a double encoder–triple decoder structure. Each decoder branch focuses on a specific category and predicts a depth map. The extra decoder surrounded by a dotted box is introduced to fuse three depth maps into one of the fusion strategies we used.

**Figure 4 sensors-23-07560-f004:**
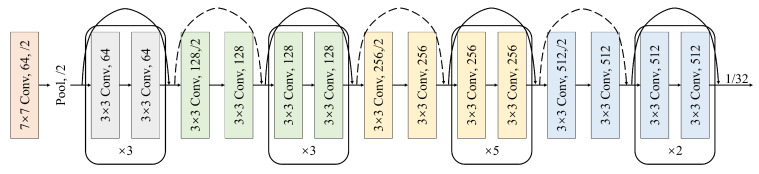
Encoder for feature extraction.

**Figure 5 sensors-23-07560-f005:**
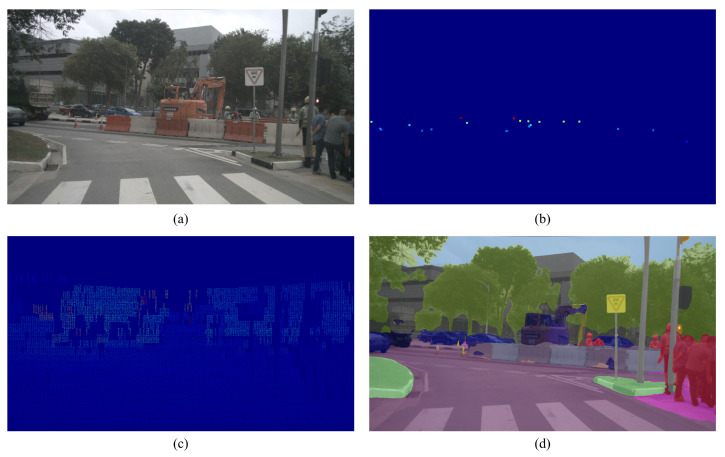
Examples of data used in the experiment. (**a**) RGB image; (**b**) millimeter-wave radar data (enhanced by 5×); (**c**) depth labels; (**d**) semantic segmentation labels.

**Figure 6 sensors-23-07560-f006:**
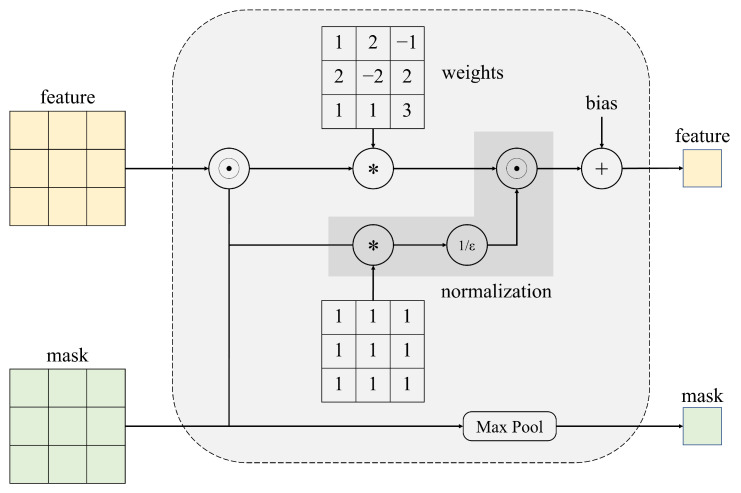
Sparse invariant convolution procedure. The symbol “*” in the image indicates a two-dimensional convolution operation, and the symbol “⊙” indicates the multiplication of the corresponding pixel positions.

**Figure 7 sensors-23-07560-f007:**
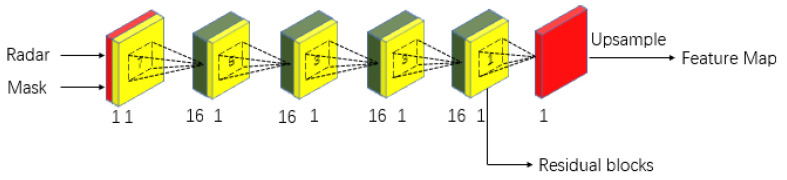
We stack 5 sparse invariant convolution layers to extract features preliminarily in radar-branch.

**Figure 8 sensors-23-07560-f008:**
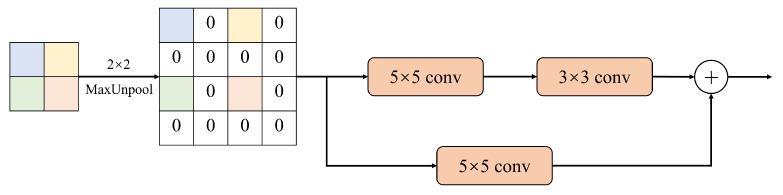
The upsampling module used in the decoder. Upsample to 2× input.

**Figure 9 sensors-23-07560-f009:**
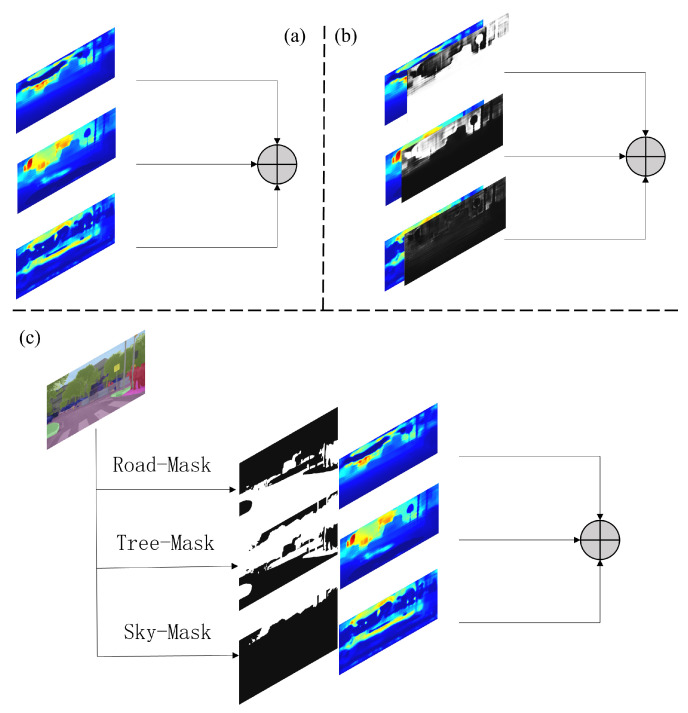
Fusion methods of depth maps. (**a**) Summation of them. (**b**) Weighed summation using confidence maps aligned with depth maps. (**c**) Select the corresponding regions using a generated semantic segmentation map.

**Figure 10 sensors-23-07560-f010:**
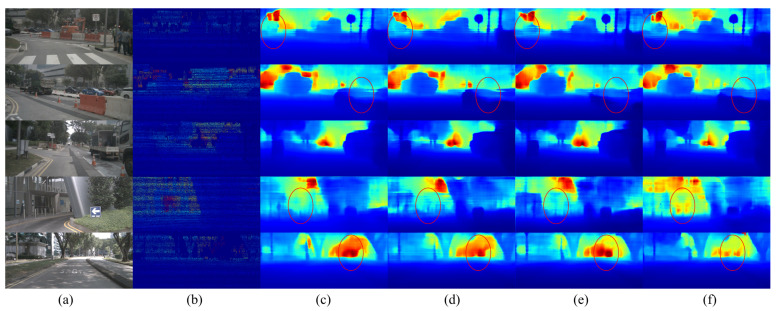
Results of depth estimation of single decoder model and our proposed method with different fusion methods on nuScenes dataset. The red circles in the figure indicate areas that require special attention.(**a**) RGB image, (**b**) ground truth, (**c**) single decoder results, (**d**–**f**) results of our proposed model using add, conf, and seg fusion methods, respectively.

**Table 1 sensors-23-07560-t001:** The results of ours with seg fusion with previous methods on nuScenes.

Method	MAE	RMSE	REL	$\delta_{1}$	$\delta_{2}$	$\delta_{3}$	Params (MB)	FPS
Ma et al. [46]	3.430	7.195	0.164	0.809	0.916	0.959	**26.107**	3.690
Hu et al. [38]	3.630	6.882	0.187	0.779	0.916	0.963	131.919	2.966
Lin et al. [47]	2.640	5.889	0.118	0.874	0.950	0.976	29.422	**22.625**
Ours with Seg	**2.424**	**5.516**	**0.112**	**0.887**	**0.956**	**0.979**	31.950	14.182

**Table 2 sensors-23-07560-t002:** The results of different values of parameters in depth loss function.

ω	Road		Tree		Sky	
	MAE	RESE	MAE	RESE	MAE	RESE
0.0	2.351	4.978	4.889	8.262	6.931	8.749
0.2	1.053	3.062	4.515	7.914	6.717	8.597
0.4	1.021	3.023	4.559	7.974	6.691	8.609
0.6	0.984	2.990	4.655	8.056	6.899	8.719
0.8	0.968	2.962	4.524	7.959	**6.331**	**8.149**
1.0	0.932	2.922	4.480	7.895	6.956	9.074
1.2	0.931	2.902	4.637	8.057	7.170	9.243
**1.4**	**0.905**	**2.853**	**4.457**	**7.845**	6.575	8.449
1.6	0.905	2.856	4.746	8.228	7.158	9.032
1.8	1.015	2.943	5.396	8.843	9.019	10.924
2.0	0.953	2.921	9.950	15.089	18.233	20.668

**Table 3 sensors-23-07560-t003:** The results of single model and different fusion methods of depth maps.

Method	MAE	RMSE	REL	$\delta_{1}$	$\delta_{2}$	$\delta_{3}$
Single	2.634	5.909	0.119	0.874	0.950	0.976
Add	2.612	5.857	0.119	0.876	0.950	0.976
Conf	2.606	5.875	0.120	0.876	0.950	0.976
Seg	**2.424**	**5.516**	**0.112**	**0.887**	**0.956**	**0.979**

**Table 4 sensors-23-07560-t004:** Depth estimation results for different distances.

Mothed	0–10 m		10–30 m		30–50 m		50–100 m	
	MAE	RESE	MAE	RESE	MAE	RESE	MAE	RESE
Single	0.582	1.581	2.343	4.296	5.996	8.164	12.340	16.604
Add	0.583	1.604	2.336	4.329	6.006	8.188	11.895	16.217
Conf	0.597	1.657	2.357	4.423	5.678	7.893	11.700	15.923
Seg	**0.559**	**1.560**	**2.203**	**4.111**	**5.595**	**7.743**	**10.910**	**15.195**

## Data Availability

Not applicable.

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
