# Peer review of "Radar-Camera Fusion Network for Depth Estimation in Structured Driving Scenes"

_sensors, 2023, doi:10.3390/s23177560_

Round 1
Reviewer 1 Report
1. Please provide the unit of evaluation metrics, MAE and RMSE.
2. What is the output of radar, i.e., x in (1)?
3. Please explain the reason that using three decoder branches for road, tree and sky respectively, not others? What will happen if only using one decoder?
4. Please give out the hyperparameters in (10) when training.
5. Please discuss that the accuracy is enough for practical application? Is there any problems?
Reviewer 2 Report
This paper proposes a novel depth estimation for autonomous driving. Overall, this work is solid and has some potential. However, some modifications should be made before publication.
(1) In the introduction, you should present the importance of the perception system in detail. In the first survey related to control system design for autonomous vehicles and connected and automated vehicles (10.1109/JIOT.2023.3307002), this article elaborates in detail on the significance of the perceptual system for downstream modules. Additionally, it also envisions the feasibility of the perception system enhancing accuracy for the estimation system. Therefore, this article needs to encompass the aforementioned work.
(2) Multi-sensor fusion is widespread for autonomous driving. For example, it could be used for vehicle state estimation: automated vehicle sideslip angle estimation considering signal measurement characteristic. Thus, you need to enumerate applications of sensor fusion in autonomous vehicles within the text, such as the fusion of speed estimation.
(3) For the comparison study, you should provide the model’s parameter size/inference time.
(4) In Figure 5, you need to zoom in on a specific area to demonstrate the effectiveness of the proposed method.
(5) Please provide the work limitations and future work direction.
